# Multi-Environment Pretraining Enables Transfer to Action Limited Datasets

## Abstract

Using massive datasets to train large-scale models has emerged as a dominant approach for broad generalization in natural language and vision applications. In reinforcement learning, however, a key challenge is that available data of sequential decision making is often not annotated with actions - for example, videos of game-play are much more available than sequences of frames paired with their logged game controls. We propose to circumvent this challenge by combining large but sparsely-annotated datasets from a *target* environment of interest with fully-annotated datasets from various other *source* environments. Our method, Action Limited PreTraining (ALPT), leverages the generalization capabilities of inverse dynamics modelling (IDM) to label missing action data in the target environment. We show that utilizing even one additional environment dataset of labelled data during IDM pretraining gives rise to substantial improvements in generating action labels for unannotated sequences. We evaluate our method on Atari game-playing environments and show that with target environment data equivalent to only 12 minutes of gameplay, we can significantly improve game performance and generalization capability compared to other approaches. Furthermore, we show that ALPT remains beneficial even when target and source environments share no common actions, highlighting the importance of pretraining on broad datasets even though they might seem irrelevant to the target task at hand.

## 1. Introduction

The training of large-scale models on large and diverse data has become a standard approach in natural language and computer vision applications (Devlin et al., 2019; Brown

---
[1]Anonymous Institution, Anonymous City, Anonymous Region, Anonymous Country. Correspondence to: Anonymous Author <anon.email@domain.com>.

et al., 2020; Mahajan et al., 2018; Zhai et al., 2021). Recently, a number of works have shown that a similar approach can be applied to tasks more often tackled by reinforcement learning (RL), such as robotics and game-playing. For example, Reed et al. (2022) suggest combining large datasets of expert behavior from a variety of RL domains in order to train a single generalist agent, while Lee et al. (2022) demonstrate a similar result but using non-expert (offline RL) data from a suite of Atari game-playing environments and using a decision transformer (DT) sequence modeling objective (Chen et al., 2021b).

Applying large-scale training necessarily relies on the ability to gather sufficiently large and diverse datasets. For RL domains, this can be a challenge, as the most easily available data – for example, videos of a human playing a video game or a human completing a predefined task – often does not contain *labelled* actions, i.e., game controls or robot joint controls. We call such datasets *action limited*, because little or none of the dataset is annotated with action information. Transferring the success of approaches like DT to such tasks is therefore bottlenecked by the ability to acquire action labels, which can be expensive and time-consuming (Zolna et al., 2020).

Some recent works have explored approaches to mitigate the issue of action limited datasets. For example, Video PreTraining (VPT) (Baker et al., 2022) proposes gathering a small amount (2k hours of video) of labeled data manually which is used to train an inverse dynamics model (IDM) (Nguyen-Tuong et al., 2008); the IDM is then used to provide action labels on a much larger video-only dataset (70k hours). This method is shown to achieve human level performance in Minecraft. It has also been demonstrated that some agents can learn directly from videos without any action labels (Seo et al., 2022).

While VPT shows promising results, it still requires over 2k hours of manually-labelled data; thus, a similar amount of expensive labelling is potentially necessary to extend VPT to other environments. In this paper, we propose an orthogonal but related approach to VPT: leveraging a large set of labeled data from various *source* domains to learn an agent policy on a limited action dataset of a *target* evaluation environment. To tackle this setting, we propose Action Limited Pretraining (ALPT), which relies on the hypothesis that

shared structures between environments can be exploited by non-causal (i.e., bidirectional) transformer IDMs. This allows us to look at both past and future frames to infer actions. In many experimental settings, the dynamics are far simpler than multi-faceted human behavior in the same setting. It has been suggested that IDMs are therefore more data efficient and this has been empirically shown (Baker et al., 2022). ALPT thus uses the multi-environment source datasets as pretraining for an IDM, which is then finetuned on the action-limited data of the target environment in order to provide labels for the unlabelled target data, which is then used for training a DT agent.

Through various experiments and ablations, we demonstrate that leveraging the generalization capabilities of IDMs is critical to the success of ALPT, as opposed to, for example, pretraining the DT model alone on the multi-environment datasets or training the IDM only on the target environment. On a benchmark game-playing environment, we show that ALPT yields as much as 5x improvement in performance, with as little as $10k$ labelled samples required (i.e., $0.01\%$ of the original labels), derived from only 12 minutes of labelled game play (Ye et al., 2021). We show that these benefits even hold when the source and target environments use distinct action spaces; i.e., the environments share similar states but *no* common actions, further demonstrating the power of IDM pretraining.

While ALPT is, algorithmically, a straightforward application of existing offline RL approaches, our results provide a new perspective on large-scale training for RL. Namely, our results suggest that the most efficient path to large-scale RL methods may be via generalist inverse dynamics modelling paired with specialized agent finetuning, instead of generalist agent training alone.

## 2. Related Work

In this section, we briefly review relevant works in multi-task RL, meta-learning for RL, semi-supervised learning, and transfer learning.

**Multi-Task RL.** It is commonly assumed that similar tasks share similar structure and properties (Caruana, 1997; Ruder, 2017; Zhang et al., 2014; Radford et al., 2019a). Many multi-task RL works leverage this assumption by learning a shared low-dimensional representation across all tasks (Calandriello et al., 2014; Borsa et al., 2016; D'Eramo et al., 2020). These methods have also been extended to tasks where the action space does not align completely (Bräm et al., 2020). Other methods assume a universal dynamics model when the reward structure is shared but dynamics are not (Zhang et al., 2021a). Multi-task RL has generally relied on a task identifier (ID) to provide contextual information, but recent methods have explored using additional

side information available in the task meta-data to establish a richer context (Sodhani et al., 2021). ALPT can be seen as multi-task RL, given that we train both the sequence model and IDM using multiple different environments, but we do not explicitly model context information or have access to task IDs.

**Meta RL.** Meta-learning is a set of approaches for *learning to learn* which leverages a set of meta-training tasks (Schmidhuber, 1987; Bengio et al., 1991), from which an agent can learn either parts of the learning algorithms (eg how to tune the learning rate) or the entire algorithm (Lacombe et al., 2021; Kalousis, 2002). In this setting, meta-learning can be used to learn policies (Duan et al., 2017; Finn et al., 2017) or dynamics models (Clavera et al., 2019). A distribution of tasks is assumed to be available for sampling, in order to provide additional contextual information to the policy. One such method models contextual information as probabilistic context variables which condition the policy (Rakelly et al., 2019). This method has been shown to learn from only a handful of trajectories. Meta-training can be used to learn policies offline, while using online interaction to correct for distribution shift, without requiring any rewards in the online data (Pong et al., 2022). These methods are commonly used to train on a source set of tasks, like ALPT, but usually require task labels. Meta-training tasks need to be hand-selected, and the results are highly dependent on the quality of that process.

**Semi-supervised learning.** *Semi-supervised* learning uses both labelled and unlabelled data to improve supervised learning performance (Zhu et al., 2009). It is especially useful when a limited amount of labelles data is given and additional labels are difficult to acquire, unlabelled data is plentiful. Early methods of this type infer unknown labels using a classifier trained on the labeled data (Zhu & Ghahramani, 2002). Other methods rely on additional structural side information to regularize supervised objectives (Szummer & Jaakkola, 2001), such as the time scale of a Markov random walk over a representation of the data. Many methods, especially those using deep learning, combine supervised and unsupervised learning objectives (Rasmus et al., 2015). More recent methods use generative models and approximate Bayesian inference to fill in missing labels (Kingma et al., 2014). The problem of semi-supervised learning is especially relevant in RL, where large datasets of experience containing action descriptions or rewards may hard to acquire, eg. through manual annotation of videos or running robotic experiments. By using an inverse dynamics model, ALPT applies semi-supervised learning to label actions in a large dataset of experience frames, given only limited labeled action data.

**Transfer Learning and Zero-shot RL.** Policies learned by RL in one domain can have limited capability to generalize

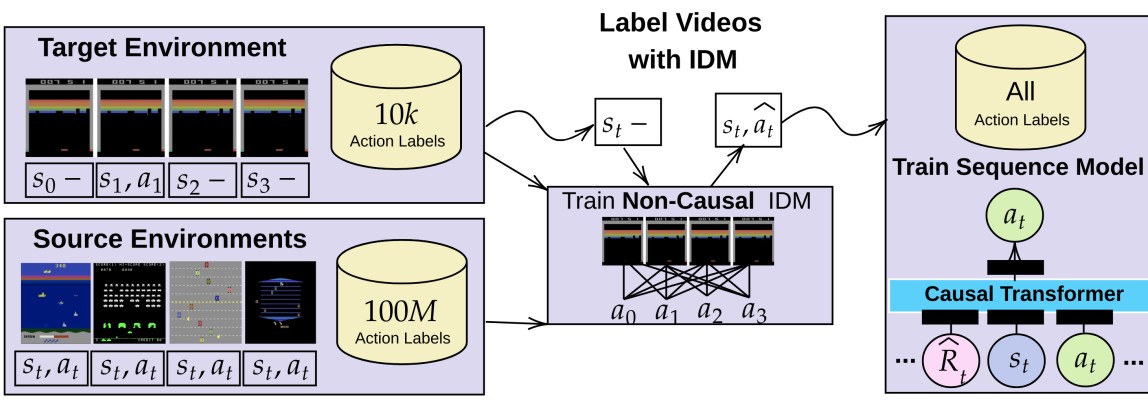

Figure 1: The dynamics model pretraining procedure of ALPT using the source set of environments along with the limited action target environment dataset.

to new settings (Oh et al., 2016). The most difficult problem is zero-shot RL, where the agent must generalize at evaluation time to a new environment that was not seen in training, without acquiring any new data. Transfer learning (Taylor & Stone, 2009) tackles a subset of generalization problems where the agent can access interactions from a related environment or task during training. This prior experience in other environments is leveraged to improve learning in novel environments. Transfer learning has been applied across both environments (Mordatch et al., 2016; Tzeng et al., 2020) and tasks (Rusu et al., 2016; Parisotto et al., 2016). It has also been examined in hard exploration games, using imitation learning from human-generated video data (Aytar et al., 2018). ALPT can be seen as tackling the transfer learning problem, with limited action data from the target environment and providing pseudo-labels for actions. Notably, we consider the under-explored scenario where the action space is not completely shared between the training and test environments.

**Offline RL.** Offline RL describes a setting that during learning, the agent has access to only a fixed dataset of experience. When function approximation becomes necessary in environments with large, complex state spaces, online RL algorithms are extremely sensitive to the dataset distribution. It has been shown that even when the logged data is collected with a single behavior policy, standard online algorithms fail (Fujimoto et al., 2019). This has been hypothesized to be due erroneous generalization of the state action function. One such class of methods applies random ensembles of Q-value function targets (Agarwal et al., 2020a). Other works suggest regularizing the agent policy to the behavior policy (Zhang et al., 2021b). In offline RL, errors arise when needed to bootstrap the value function and algorithms must apply strong regularizations on both learned policy and value function to achieve stable performance (Wu et al., 2020; Kumar et al., 2019; Zhang et al.,

2021b; Nachum et al., 2019). We use decision transformers (DT) (Chen et al., 2021b) as the backbone for ALPT, and these have been shown to be successful in learning generalizable agents from logged data in a diverse set of environments (Lee et al., 2022).

## 3. Background

In this section, we review the standard offline RL setting and the use of decision transformers (DT) as a sequence modelling objective for offline RL. We then define the setting of multi-environment offline RL with action-limited data, which is our focus.

### 3.1. Offline Reinforcement Learning

We consider an agent acting within a Markov decision process (MDP) defined by $\langle \mathcal{S}, \mathcal{A}, \mathcal{P}, \mathcal{R} \rangle$, where $\mathcal{S}$ is the set of states, $\mathcal{A}$ is the set of actions, $\mathcal{P} : \mathcal{S} \times \mathcal{A} \to \text{Dist}(\mathcal{S})$ is the transition probability kernel and $\mathcal{R} : \mathcal{S} \times \mathcal{A} \to [0, 1]$ is the scalar reward function.

In offline RL, the agent is given a dataset of episodes, i.e., sequences of states, actions, and rewards collected by unknown policies interacting with the environment:

$$\langle \dots, \mathbf{s_t}, a_t, r_t, \dots \rangle. \tag{1}$$

The objective is typically to use this dataset in order to learn a conditional action distribution, $P_\theta(a_t | \mathbf{s}_{\leq t}, a_{<t}, r_{<t})$, that maximizes the expectation of the total return, $G_t = \sum_{k \geq 0} r_{t+k}$ when used to interact with the environment from which the training episodes were generated.

### 3.2. Offline RL as Sequence Modeling

Decision transformer (DT) (Chen et al., 2021a) is an approach to offline RL which formulates this problem as sequence modeling, and then uses transformer-based architec-

tures to solve it. For this purpose, the episodes in the offline dataset are augmented with the returns associated with each step:

$$\tau = \langle \dots, \mathbf{s_t}, a_t, r_t, G_t, \dots \rangle. \qquad (2)$$

This sequence is tokenized and passed to a causal transformer $P_\theta$, which predicts both returns and actions using a cross-entropy loss. Thus, the learning objective for $\theta$ is:

$$
\begin{aligned}
J(\theta) = \mathbb{E}_\tau \Big[ \sum_t & -\log P_\theta(G_t | \mathbf{s}_{\leq t}, a_{<t}, r_{<t}) \\
& - \log P_\theta(a_t | \mathbf{s}_{\leq t}, a_{<t}, r_{<t}, G_t) \Big].
\end{aligned} \qquad (3)
$$

During inference, at each timestep $t$, after observing $\mathbf{s}_t$, DT uses the predicted return distribution $P_\theta(G_t | \mathbf{s}_{\leq t}, a_{<t}, r_{<t})$ to choose an optimistic estimate $\hat{G}_t$ of return, before using $P_\theta(a_t | \mathbf{s}_{\leq t}, a_{<t}, r_{<t}, \hat{G}_t)$ to select an action $\hat{a}_t$ (see Lee et al. (2022) for details).

### 3.3. Multi-Environment and Action Limited Datasets

Our goal is to use pretraining on a set of environments where labelled data is plentiful, in order to do well on a target environment where only limited action-labelled data is available. Therefore, the offline RL setting we consider includes multiple environments and action-limited datasets, as we detail below.

We consider a set of $n$ *source* environments, defined by a set of MDPs: $E = \{\mathcal{M}_1, \dots, \mathcal{M}_n\}$, and a single *target* environment $\mathcal{M}_\star$. For each source environment $\mathcal{M}_d$, we have an offline dataset of episodes generated from $\mathcal{M}_d$, denoted by $\mathcal{D}_d = \{\tau := \langle \dots, \mathbf{s}_t, a_t, r_t, \dots \rangle\}$, fully labelled with actions. For the target environment, the agent has access to a small labelled dataset from $\mathcal{M}_\star$, denoted as $\mathcal{D}_\star^+ = \{\tau := \langle \dots, \mathbf{s}_t, a_t, r_t, \dots \rangle\}$, and a large dataset without action labels, $\mathcal{D}_\star^- = \{\tau := \langle \dots, \mathbf{s}_t, r_t, \dots \rangle\}$.

## 4. Action Limited Pretraining (ALPT)

We now describe our proposed approach to offline RL in multi-environment and action limited settings. ALPT relies upon an inverse dynamics model (IDM) which uses the combined labelled data in order to learn a representation that generalizes well to the limited action data from the target environment. The predicted labels of the IDM on the unlabelled portion of the target environment dataset are then used for training a sequence model parameterized as a decision transformer (DT). We elaborate on this procedure below and summarize the full algorithm in Table 1.

### 4.1. Inverse Dynamics Modeling

Our inverse dynamics model (IDM) is a bidirectional transformer trained to predict actions from an action-unlabelled sub-trajectory of an episode. The training objective for learning an IDM $P_\beta$ is

$$J(\beta) = \mathbb{E}_\tau \left[ \sum_t \sum_{i=0}^{k-1} -\log P_\beta(a_{t+i} | \mathbf{s}_t, \dots, \mathbf{s}_{t+k}) \right], \quad (4)$$

where $k$ is the length of training sub-trajectories. In our experiments, we use $k = 5$ and parameterize $P_\beta$ using the GPT-2 transformer architecture (Radford et al., 2019b), modified to be bidirectional by changing the attention mask.

### 4.2. Multi-Environment Pretraining and Finetuning

ALPT is composed of a two-stage **pretraining** and **finetuning** process.

During **pretraining**, we use the combined labelled datasets for all source environments combined with the labelled portion of the target environment dataset: $(\bigcup_{d=1}^n \mathcal{D}_d) \cup \mathcal{D}_\star^+$, to train the IDM $P_\beta$. Concurrently, we also train the DT $P_\theta$ on the combined labelled and unlabelled datasets for all source environments combined with the target environment datasets, by using the IDM to provide action labels on the unlabelled portion $\mathcal{D}_\star^-$. The DT training dataset is therefore: $(\bigcup_{d=1}^n \mathcal{D}_d) \cup \mathcal{D}_\star^+ \cup \mathcal{D}_\star^-$.

During **finetuning**, we simultaneously train both the IDM and DT exclusively on the target environment dataset. We train the IDM on the labelled portion $\mathcal{D}_\star^+$. We train DT on the full action limited dataset $\mathcal{D}_\star^+ \cup \mathcal{D}_\star^-$ by using the IDM to provide action labels on the unlabelled portion $\mathcal{D}_\star^-$.

Finally, during evaluation we use the trained DT agent to select actions in the target environment $\mathcal{M}_\star$, following the same protocol described in Section 3.2.

## 5. Experiments

We evaluate ALPT on a multi-game Atari setup similar to Lee et al. (2022). Our findings are three-fold: (1) ALPT, when pretrained on multiple source games, demonstrates significant benefits on the target game with limited action labels; (2) ALPT maintains its significant benefits even when pretrained on just a single source game with a disjoint action space, (3) we demonstrate similar benefits on maze navigation tasks.

### 5.1. Experimental Procedure

**Architecture and Training.** Our architecture and training protocol follow the multi-game Atari setting outlined in Lee et al. (2022). Specifically, we use a transformer with 6 layers of 8 heads each and hidden size 512. The rest of the architecture and training hyperparameters remain unchanged for experiments on Atari. For the Maze navigation experiments, we modify the original hyperparameters to use a batch size of 256 and a weight decay of $5 \times 10^{-5}$. During

Table 1: A summary of ALPT.

| Step | Procedure |
|---|---|
| **Pretraining** | Train IDM on all labelled data: $(\bigcup_{d=1}^{n} \mathcal{D}_d) \cup \mathcal{D}_\star^+$. |
| | Train DT on all data: $(\bigcup_{d=1}^{n} \mathcal{D}_d) \cup \mathcal{D}_\star^+ \cup \mathcal{D}_\star^-$, with IDM providing action labels on $\mathcal{D}_\star^-$ |
| **Finetuning** | Train IDM on labelled data in target environment dataset: $\mathcal{D}_\star^+$. |
| | Train DT on all data in target environment dataset: $\mathcal{D}_\star^+ \cup \mathcal{D}_\star^-$, with IDM providing action labels on $\mathcal{D}_\star^-$ |
| **Evaluation** | Use trained DT agent to interact with target environment $\mathcal{M}_\star$. |

pre-training, we train the DT and IDM for $1M$ frames. The details of all parameters can be found in Appendix B.

**Datasets**  As in Lee et al. (2022), we use the standard offline RL Atari datasets from RL Unplugged (Gulcehre et al., 2020). Each game's dataset consists of 100M environment steps of training a DQN agent (Agarwal et al., 2020b). For the source games, we use this dataset in its entirety. For the target game, we derive an action-limited dataset by keeping the action labels for randomly sampled sequences consisting of a total 10k transitions (0.01% of the original dataset, equivalent to 12 minutes of gameplay) and removing action labels in the remainder of the dataset. For the maze navigation experiments, we generate the data ourselves. The offline datasets for each maze configuration contain 500 trajectories with a length of 500 steps or until the goal state is reached. They are generated using an optimal policy for each maze, with an $\epsilon$-greedy exploration rate of 0.5 to increase data diversity.

### 5.2. Baseline Methods

We detail the methods that we compare ALPT to below.

**Single-game variants.** To evaluate the benefit of multi- versus single-environment training, we assess the performance of training either DT alone or DT and IDM simultaneously on the target game. When training DT alone (**DT1**), we train it only on the 10k subset of data that is labelled, while when training DT and IDM simultaneously (**DT1-IDM**) we use the IDM to provide action labels on the unlabelled portion of the data.

**Multi-game DT variants.**  To assess the need for IDM versus training on DT alone, we evaluate a multi-game baseline (**DT5**) composed of DT alone. For this baseline, we pretrain DT on all labelled datasets combined from both the source and target environments before finetuning the DT model on the 10k labelled portion of the target game.

**Return prediction DT variants.** As an alternative way for DT to leverage the unlabelled portion of the target environ-

Table 2: A summary of the baseline methods.

| Method | Training Games | IDM |
|---|---|---|
| DT-1 | 1 | × |
| DT-5 | 5 | × |
| DT-1-IDM | 1 | ✓ |
| DT-5-RET | 1 | × |
| **ALPT-**$X$ | $X$ if specified, otherwise 5 | ✓ |

ment dataset, we evaluate a baseline (**DT5-RET**) that uses the unlabelled portion for training its return prediction. The model still undergoes a pretraining and finetuning stage, first pretraining on all available data and then finetuning only on data from the target game.

We give a summary of the baseline methods as well as ALPT in Table 2. We also present results of an additional variant of ALPT in which only the IDM is pretrained (rather than both the IDM and DT) in Appendix A.

### 5.3. How does ALPT perform compared to the baselines?

We focus our first set of multi-game pretraining experiments on 5 Atari games: {*Asterix, Breakout, SpaceInvaders, Freeway, Seaquest*}. This subset of games is selected due to having a similar shared game structure and access to high-quality and diverse pretraining data. We evaluate each choice of target game in this setting, i.e., for each game we evaluate using it as the target game while the remaining 4 games comprise the source environments. We compare our pretraining regime (ALPT) with the single-game variant and standard DT baselines in Figure 2. We see that pretraining ALPT on the source games results in substantial downstream performance improvements. We show that there are relatively minimal performance improvements when pretraining on datasets that do not include any non-target environments (DT1-IDM). Utilizing ALPT results in improvements up to $\approx 500\%$ higher than the single-game training regime. The performance difference is especially stark in *Breakout* and

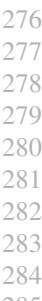
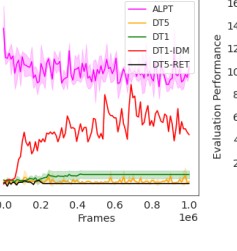
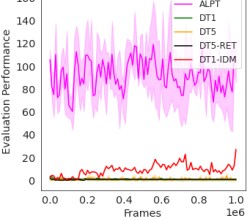
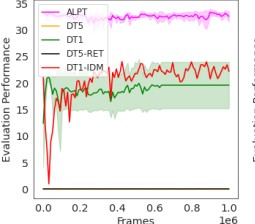
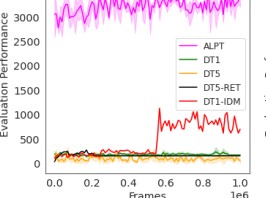
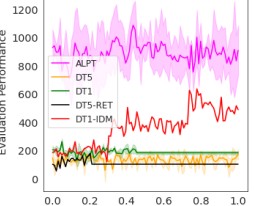

Figure 2: Game performance across the ALE environments for the baseline and ALPT. The figure shows the evaluation game performance (Episodic Return) of our DT policies during finetuning on the limited action target dataset. Higher score is better. The shaded area represents the standard deviation over 3 random seeds. The $x$-axis shows the number of finetuning steps. We evaluate ALPT on 16 episodes of length 2500 each following (Lee et al., 2022).

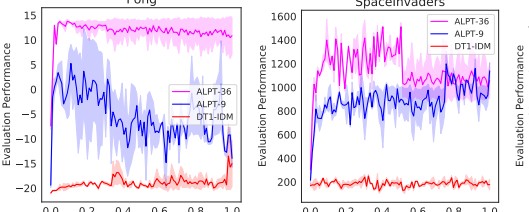
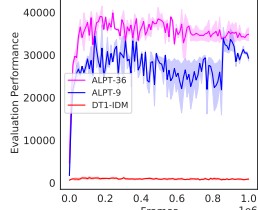
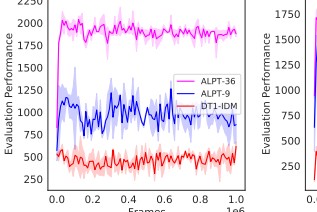
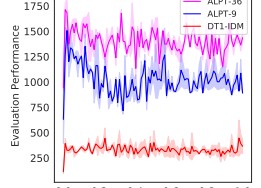

Figure 3: We evaluate performance of ALPT with a higher number of source games. We show performance of ALPT trained on 36 Atari source games (ALPT-36) and ALPT trained on 9 Atari source games (ALPT-9). We find that performance generally improves with more source games.

*Seaquest*. Additionally, we show that performance is not recovered under pretraining only the sequence model (DT5) on the action rich environments, indicating that most generalization benefits are occurring due to the IDM pretraining. We also show that performance using DT1 and DT5-RET is poor, highlighting the need for explicit re-labelling to achieve good performance during sequence model finetuning.

We also encourage the reader to look to Appendix A for a comparison to a variant of ALPT for which only the IDM is pretrained, while the DT is initialized from scratch during finetuning. We find that this variant maintains strong performance compared to DT1-IDM, suggesting that the main benefit of pretraining is the IDM.

For completeness, we examine the performance of a standard offline RL algorithm (Conservative Q-Learning) (Kumar et al., 2020) against ALPT in Appendix C.

### 5.4. Is dynamics modelling improved under more or less source datasets?

In this set of experiments, we expand the set of source and target games. As in the previous experiments, each source game provides a fully-labelled dataset of size 100M, while each target game has a dataset of size 100M with only 10k action labels. We evaluate performance using

5 target games ({*'Pong', 'SpaceInvaders', 'StarGunner', 'MsPacman', 'Alien'*}) and using either 36 source games (ALPT-36, trained on all 41 game datasets available in Agarwal et al. (2020b) minus the 5 target games) and using 9 source games (ALPT-9, trained on {*'Asterix', 'Breakout', 'Freeway', 'Seaquest', 'Atlantis', 'DemonAttack', 'Frostbite', 'Gopher', 'TimePilot'*}). Note that for each of these ALPT variants, we perform a *single* pretraining phase and then *multiple* finetuning phases (one for each target game), thus showing that a single pretrained model can be transferred to various target games.

We compare pretraining with ALPT to training DT1-IDM on each target game alone. Results are presented in Figure 3, and show that target game performance improves with more source games.

### 5.5. Can ALPT help when source and target have disjoint action spaces?

The previous experiments have included at least one source game during pretraining whose the action space overlaps with that of the target environment. The next set of experiments aims to explore pretraining on source environments where the action space ($\mathcal{A}_d$) is disjoint with the target environment action space ($\mathcal{A}_*$), that is, $\mathcal{A}_d \cap \mathcal{A}_* = \emptyset$. To do so, we use *Freeway* as our single source environment dataset. In *Freeway*, the action space consists of {Up, Down}. In

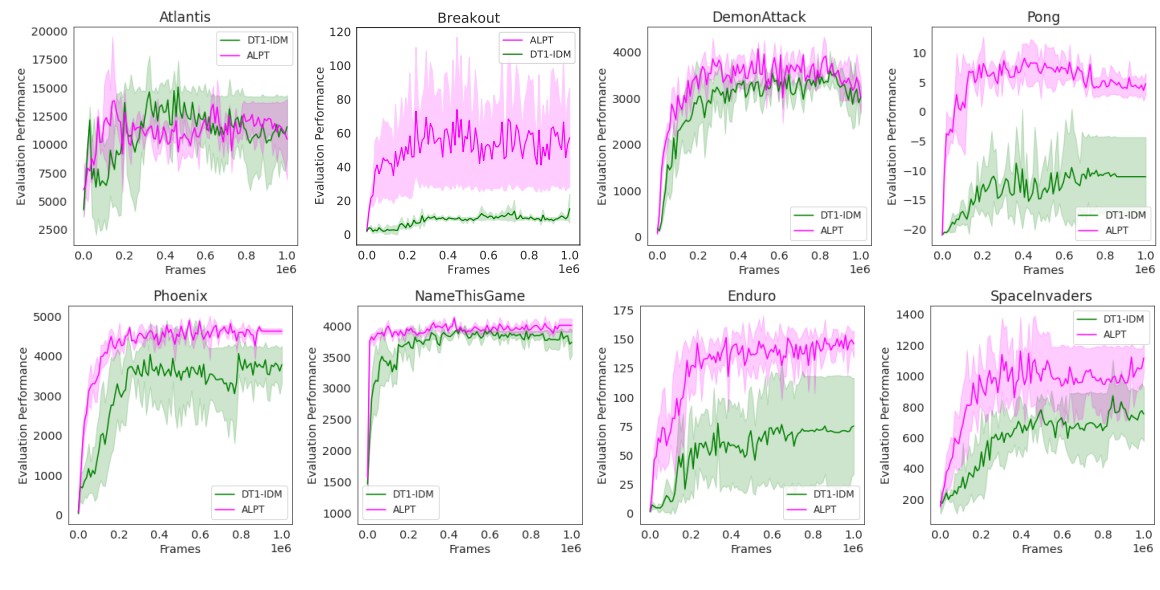

Figure 4: We evaluate performance of ALPT when source and target games have disjoint action spaces. In each of these plots we pretrain using a single source game Freeway. Despite the disjoint action space, we still see benefits of pretraining.

contrast, a game such as *Breakout* has an action space consisting of {Left, Right}. Surprisingly, using *Freeway* as a source environment and *Breakout* as a target environment still yields significant benefits for ALPT. We present this result in Figure 4, as well as a variety of other choices for the target environment, none of which share any actions with the source environment *Freeway*.

We hypothesize that performance improvements are not unexpected due to the already known broad generalization capabilities of transformer architectures. It may also be the case that, despite the action spaces being disjoint, they still exhibit similar structure. For example, a top-down action space and a left-right action space are structurally opposite to each other in terms of movement, differing only in orientation. This similar structure is potentially learned and leveraged during finetuning.

### 5.6. Do ALPT's benefits persist in other domains ?

We now demonstrate ALPT's benefits on an action-limited navigation task. This corresponds to a scenario where we have densely annotated navigation maps for a set of source regions but only a sparsely annotated navigation map for a target region. We would like to evaluate whether ALPT, pretrained on source regions with abundant action labels, can generalize to navigating in a target region (of a different layout) with limited action labels.

**Maze Navigation Environments.** To answer the above question, we consider a gridworld navigation task where an agent seeks to navigate to a goal location in a $20 \times 20$ 2D maze from a random starting location to a random goal

location using 4 discrete actions: {Up, Down, Left, Right}. The agent receives a reward of 1 at the goal state and $r = 0$ otherwise. To collect the offline training datasets, we follow Yang et al. (2022); Zhang et al. (2018) to algorithmically generate maze layouts with random internal walls that form blocked or tunneled obstacles as shown in Figure 5. We start with blocked obstacles, and generate one *source* maze from which we collect 500 trajectories with full action labels. We then use a different random seed to generate the *target* maze, from which we collect 500 trajectories with only 250 action labels ($0.5\%$ of the full action labels). We then train the IDM of ALPT-Blocked on both the source and target datasets, labeling the missing actions from the target game, and train DT all at the same time (no separate finetuning stage). ALPT-Blocked and other baselines are evaluated in the target maze only.

**Results.** Performance in the target maze environment with limited action labels is presented in Figure 5 (b). ALPT-Blocked trained on both source and target mazes allows us to solve the target task twice as fast compared to only training on the target maze without access to another source maze. To further illustrate the benefit of multi-environment training on more diverse data, we introduce the tunnelled maze, and train ALPT-Blocked+Tunnelled on 500 trajectories with full actions from a source blocked maze and a source tunneled maze, respectively, as well as 250 action samples of the target blocked maze. Training on both tunnelled and blocked mazes enables greater dataset diversity, which further improves generalization, leading to even faster convergence on the target task. These preliminary results on navigation suggest that multi-environment pretraining can

benefit a broad set of tasks.

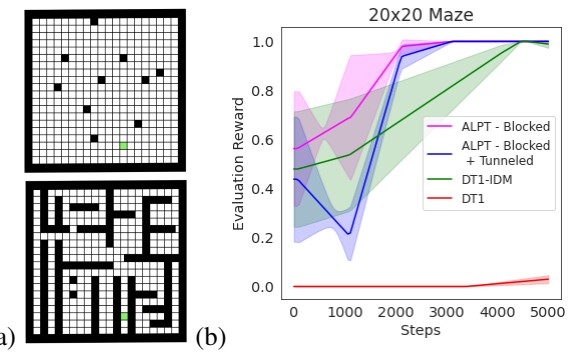

Figure 5: (a) An example diagram of the Blocked (above) and Tunneled (below) mazes. The green cell is the goal state. (b) Evaluation performance while training on the 20x20 Maze dataset. Higher score is better. The shaded area represents the standard deviation over 3 random seeds. The action limited target dataset contains 250 labelled actions in these experiments.

## 6. Conclusion

We explored the problem of learning agent policies from action limited datasets. Inspired by the paradigm of large-scale, multi-environment training, we proposed ALPT, which pretrains an inverse dynamics model (IDM) on multiple environments to provide accurate action labels for a decision transformer (DT) agent on an action limited target environment dataset. Our experiments and ablations highlight the importance of pretraining the IDM, as opposed to pretraining the DT agent alone. Our results support the importance of generalist inverse dynamics models as an efficient way to implement large-scale RL. As more labelled data becomes available for training offline RL agents, ALPT provides an efficient way of bootstrapping performance on new tasks with action limited data.

### 6.1. Limitations

The largest limitation of ALPT is the assumption that we would have plentiful labelled data from related environments. One interesting fact we uncover is that this data does not have to be based on the same action space as the desired target environment, indicating the versatility of ALPT and its ability to ingest diverse source environment training data. We also caution that we have only evaluated ALPT so far on limited, self-contained video game tasks and simple navigation environments. We hope that as more labelled data becomes available in RL domains, ALPT will have wider applicability, allowing RL agents to scale and bootstrap to new environments. It would also be useful to investigate further how much labelled data from a limited set of source

environments is required to be able to handle a much larger set of unlabelled datasets.

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

## A. Experiments with no DT pretraining

In the following set of experiments, we pretrain only the IDM component of ALPT and not the DT. We show the finetuning performance results for the Narrow set of Atari games in Figure 6. Note that the axis here is up to 100k steps as opposed to 1M for the figures in the main text.

## B. Implementation Details

In Table 3 we give the implementation details of our IDM and DT transformer architectures.

The IDM model is the same as the DT model, except that it is non-causal. This is enforced by changing the attention mask to a matrix of all 1 values in the IDM.

Table 3: A summary of the transformer model parameters.

| Parameter | Value |
|---|---|
| Layers | 6 |
| Hidden Size | 512 |
| Heads | 8 |
| Batch Size | 256 |
| Weight Decay | $5 \times 10^{-5}$ |
| Learning Rate | $3 \times 10^{-4}$ |
| Gradient Clipping | 1.0 |
| $\beta_1, \beta_2$ | 0.9, 0.999 |
| Warm-up Steps | 4000 |
| Optimizer | LAMB |

## C. Experiments with Conservative Q-Learning (CQL)

In this set of experiments, we examine the performance of Conservative Q-Learning (CQL) (Kumar et al., 2020) trained on a dataset of $10,000$ frames, as opposed to $500,000$ in the original work (Table 3 of CQL, $1\%$ dataset size), from various Atari games utilized in our experiments. In Table 4 we report the final evaluation performance on the game after training for 100 iterations. All implementation details are consistent with the original implementation in the cited work. We utilize the CQL($\mathcal{H}$) method.

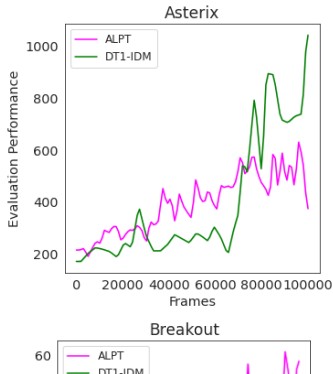

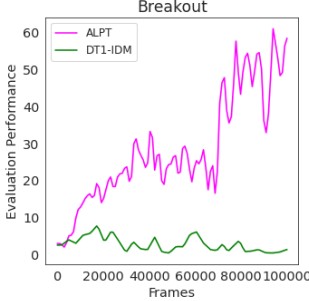

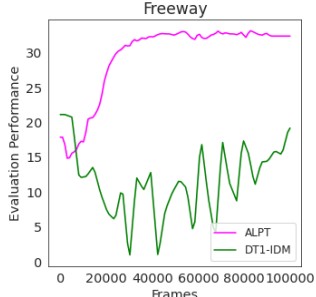

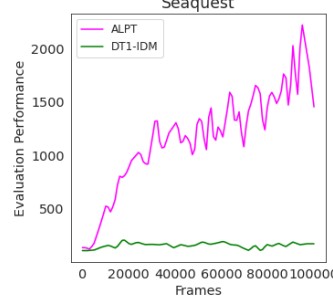

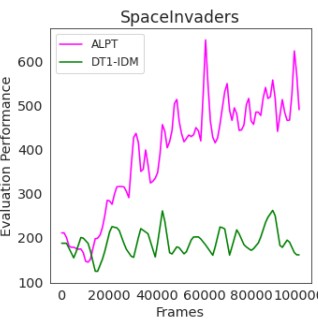

Figure 6: Evaluation game performance during finetuning of ALPT and DT1-IDM. In these experiments we do not pretrain the DT. $100k$ steps are shown.

Table 4: The final evaluation game performance after training CQL for 100 iterations on a dataset of 10,000 labelled frames from each Atari game.

| Game Name | Final Performance |
|---|---|
| Asterix | 227.5 |
| Breakout | 12.3 |
| Freeway | 10.2 |
| Seaquest | 236.0 |
| SpaceInvaders | 250.9 |

## D. Source Code

We make the source code publicly available for our Maze experiment only at this time. The details can be found at: https://anonymous.4open.science/r/alpt_maze-5927/README.md.