# OpenReview forum: "Multi-Environment Pretraining Enables Transfer to Action Limited Datasets"
_ICLR.cc/2023/Workshop/RRL — RRL 2023 Poster_

### Official Review · Reviewer_wqLj · 2023-02-28
**A good paper, but the weak points need further improvement**

**Rating:** 3
**Confidence:** 4

**Review:**

Summary:
The authors address the problem of offline RL learning from action limited datasets. They propose to pretrain an inverse dynamics model on multiple environments to provide action labels for a decision transformer. They demonstrate the performance of ALPT by comparing their algorithm to a few baselines on Atari games. Their results support the importance of generalist inverse dynamics models as an efficient approach to large-scale RL.

Strong points:
Overall, I really liked this paper. The proposed method is simple, the motivations and intuitions are well explained. I will list the main strong points:
- In many real scenarios, obtaining manually labelled data could be extremely expensive or even impossible. This paper proposes to overcome this challenge by combining large but sparsely-annotated datasets from a target environment of interest with fully-annotated datasets from other source environments.
- The paper is clearly written and well organized.
- The related work is relevant and seems complete.
- The experiments are overall well designed. I found the empirical study to be very thorough and comprehensive, including multiple baselines and ablations. In addition, the authors support their claims regarding the generalization issues with well-designed experiments and metrics.

Weak points:
I will now list a few weak points of the paper：

- In Section 5.5 and Section 5.6, the disjoint action example is too simple. Regarding generalization, more complex application scenarios can be given.
- 3 seeds is a very small number. I know that very few works present much more than this, but I think it is bad practice. Please consider adding more (>=10).

---

### Official Review · Reviewer_LdAi · 2023-03-01
**Well-illustrated work on pretraining for action-limited data**

**Rating:** 4
**Confidence:** 3

**Review:**

The paper is focused on how pretraining with labeled action data benefits acting in unannotated environments. Experiments show that the result is promising compared to the baseline. In this review, we list the strengths and weaknesses of the work.

The topic fits the guideline of Reincarcinating RL. The background and the related work parts are detailed and help illustrate the innovative insight of the work. The method part can be more specific and there can be some intuitive explanations why ALPT works well. The experiments comparing ALPT with the baseline involve detailed illustrations, but the maze navigation domain results can be expanded.

In summary, ALPT works well in the two domains mentioned in the paper and illustrates the importance of IDM pretraining and its efficiency on action-limited data.